# Effectiveness and user experiences of a valgus brace in patients with knee osteoarthritis: A mixed-method randomised controlled trial

**Lex D. de Jong**[1], **Babette C. van der Zwaard**[2], **Matthijs Y. H. van Blommestein**[1], **Corné J.M. van Loon**[1]*

1 Department of Orthopaedics, Rijnstate Hospital, Arnhem, The Netherlands, 2 Department of Orthopaedics, Jeroen Bosch Hospital, 's-Hertogenbosch, The Netherlands

* CvanLoon@rijnstate.nl (CJMvL)

## Abstract

### Background

Patients with medial compartment knee osteoarthritis (OA) may benefit from wearing a valgus brace, but previous studies lacked consideration for brace wearing time, co-interventions and the brace users' perceptions. This mixed-method randomised controlled trial investigated the effectiveness of a valgus brace on knee pain and activity limitations alongside exploring user perceptions.

### Materials and methods

Participants randomised to the intervention group (n = 23) received regular care and a customisable valgus brace for 6 months while the participants randomised to the wait-list control group (n = 23) were allowed any regular care treatment except for a knee brace. Outcomes were evaluated with a multilevel linear regression analysis at base-line, 2 weeks, 3 months and 6 months. The primary outcome was knee pain intensity at 6 months measured by a 10-cm Visual Analogue Scale (VAS). Secondary outcomes included walking distance, generic health status, knee functioning and satisfaction with the brace. Qualitative interviews were conducted with a subsample of intervention group participants and transcripts were analysed using deductive thematic analysis.

### Results

After 6 months, only a statistically significant and clinically important difference of 2.13 cm (95% CI −3.57 to −0.69) on the VAS score for knee pain intensity after a walk test was found between the intervention and control groups. Although the qualitative findings echoed some participants' negative or mixed feelings about the brace, several participants perceived positive changes in pain, other body functions, during activities and had positive user experiences.

**Data availability statement:** All relevant data are within the manuscript and its Supporting Information files. All individual participant data are collated in the S1 File which can be found here: https://figshare.com/s/50cace-e63c01e2c2f8a6. An overview of all themes, subthemes and the original verbatim Dutch and their English clean verbatim translations of the quotes emerging from the interviews can be found in S1 Table (https://figshare.com/s/20212f8bb7fba462d17f). A comprehensive triangulation of quantitative outcomes and qualitative (sub)themes in overlapping areas of data is provided in S2 Table (https://figshare.com/s/7c3894a86e4b00872d41)

**Funding:** To conduct the study, Stichting Rijnstate Ziekenhuis received financial support from Bauerfeind AG (Zeulenroba-Triebes, Germany; https://www.bauerfeind.de/de; study code BF18-OR-02). Bauerfeind AG also provided the study SecuTec® OA braces free of charge. The funder had no role in study design, data collection and analysis, decision to publish, or preparation of the manuscript.

**Competing interests:** The authors have declared that no competing interests exist.

## Conclusion

Using a valgus brace decreased knee pain intensity during walking activities. Although other outcomes showed limited effectiveness, the study's underpowered nature increased the risk of type II errors. Qualitative data highlighted positive user experiences, suggesting potential benefits beyond the measured quantitative outcomes.

## Introduction

The knee is one of the joints mostly affected by osteoarthritis (OA) [1]. Knee OA is most prevalent in women and increases with age and body weight [2–5]. As populations are ageing and becoming more obese, the incidence and prevalence of knee OA are also rising worldwide [6,7]. Knee OA can cause pain and functional limitations [8].

One common treatment option for knee OA is total knee arthroplasty (TKA) or unicompartmental knee arthroplasty (UKA). However, these treatment options are usually only considered for older patients with end-stage OA and for whom conservative options have been exhausted [9]. For younger patients or for those who have lower OA grades, it is generally accepted that high tibial osteotomy [10] or conservative treatment options are tried first [11]. The latter include intra-articular injections [12,13], exercise [14,15] or a combination thereof [16,17]. While the results of some studies suggest that assistive devices such as knee orthoses or braces can also offer some patients knee pain relief [18], the results of systematic reviews have also shown that evidence regarding the effectiveness of knee braces in general was of low quality or inconclusive at best [19,20].

The medial tibiofemoral compartment is primarily affected in most patients with knee OA [21]. This is commonly accompanied by varus deformity of the knee. This deformity can increase the risk of disease progression [22] and cartilage wear through medial compartment overload [23]. The results of almost all studies investigating the effects of braces that can correct varus deformity have suggested that these valgus braces can positively influence knee pain, knee function and the patients' activity levels. However, the results of systematic reviews and meta-analyses [24,25] showed that the limited number of identified randomised controlled trials (RCTs) were judged to have moderate levels of evidence. The majority of these RCTs [e.g., [26–32] did also not consider, control for, or report brace-wearing time or confounding co-interventions such as the concurrent use of pain medication, intra-articular injections and leg exercises. Most non-randomised studies only investigated effects on pain and activity levels up to three months of brace use. To our knowledge, there is also a dearth of qualitative studies evaluating orthotic and assistive device use in general and knee braces in particular. This limits a deeper understanding of the interplay between quantitative and qualitative patient outcomes, and this may not be conducive to driving advancements in knee brace design.

Considering the various limitations of previous valgus brace studies, and aiming for findings with high validity, applicability and trustworthiness [33], we set out to conduct a mixed-method study to comprehensively explore the effectiveness and user experiences of a valgus brace for the treatment of medial compartment knee osteoarthritis.

## Materials and methods

### Design and participants

This was a single-centre randomised clinical trial with a mixed-method (explanatory sequential) design. A convenience sample of patients from an outpatient orthopaedic department of a tertiary non-academic hospital in The Netherlands was recruited. Patients with any level of pain in the medial knee compartment were eligible for participation if they were between 40–75 years of age, had unilateral radiographically confirmed OA (Kellgren & Lawrence classification grade 2 and 3 [34]) and had varus leg alignment >0° [35]. Patients treated with intra-articular injections with glucocorticosteroids combined with analgesics in the knee were considered for participation only if the injection was received more than three months before the baseline assessment. Patients were not eligible for participation if they had i) patellofemoral osteoarthritis, ii) any symptomatic pathology (e.g., back pain, ankle pain) or systemic disease (e.g., rheumatoid arthritis, fibromyalgia) that negatively impacted their musculoskeletal system or judged to potentially confound the study outcomes, iii) pre-existing skin problems around the knee, iv) a Body Mass Index (BMI) > 35, v) physical or mental impairments as judged by the patients' own orthopedic surgeon or vi) a language barrier that prevented them from following instructions to adequately use the knee brace.

Prior to starting the study, ethical approval was granted by METC Oost-Nederland (NL66797.091.18). The study was prospectively registered in the Netherlands Trial Register on August 22, 2018 (NTR7441/NL7242/ABRnr: 66797). Written informed consent was obtained from all participants.

### Sample size and randomisation

Sample size calculation was based on a baseline Visual Analogue Scale (VAS) score of 60 mm (standard deviation 22 mm) in patients with knee OA [36], and a minimally clinically important difference (MCID) of 15 mm on the VAS [37,38]. With alpha set at 5% (two-sided) and beta at 80%, the required number of participants was 34 per group. Anticipating a 15% dropout rate we aimed to recruit 80 participants.

For logistical reasons (explained in the legend of Fig 1), potential participants had to be randomised prior to their baseline visit. Using a computer-generated randomisation list, potential participants were randomised in five blocks of 16 (1:1 allocation ratio). Group allocation was revealed to the participants before the baseline assessment. Neither the participants nor the assessor were blinded to group allocation.

### Brace (intervention) group

After baseline assessment, each participant allocated to the intervention group was fitted with a customisable knee brace (SecuTec® OA, Bauerfeind, please see https://www.bauerfeind-group.com/en/products/orthoses/knee/details/product/secutec-oa for product and technical details) by an independent, experienced orthotist. The brace was individually fitted and adjusted to produce a valgus force about the knee via its rigid lateral frame and adjustable joint system, which apply a laterally directed force to the lower leg relative to the thigh. This counteracts the knee's natural varus alignment in medial knee osteoarthritis, thereby distracting and unloading the medial compartment by shifting the weight-bearing axis—and thus the load—towards the lateral compartment through a reduction in the knee adduction moment [39]. All participants were allowed to continue their regular care and recommended to increase the brace-wearing time by about one to two hours per day in the first two weeks and to wear the brace as much as possible during activities in the remaining 22 weeks. However, wearing time was always at the participants' discretion. Participants kept diaries to self-report their

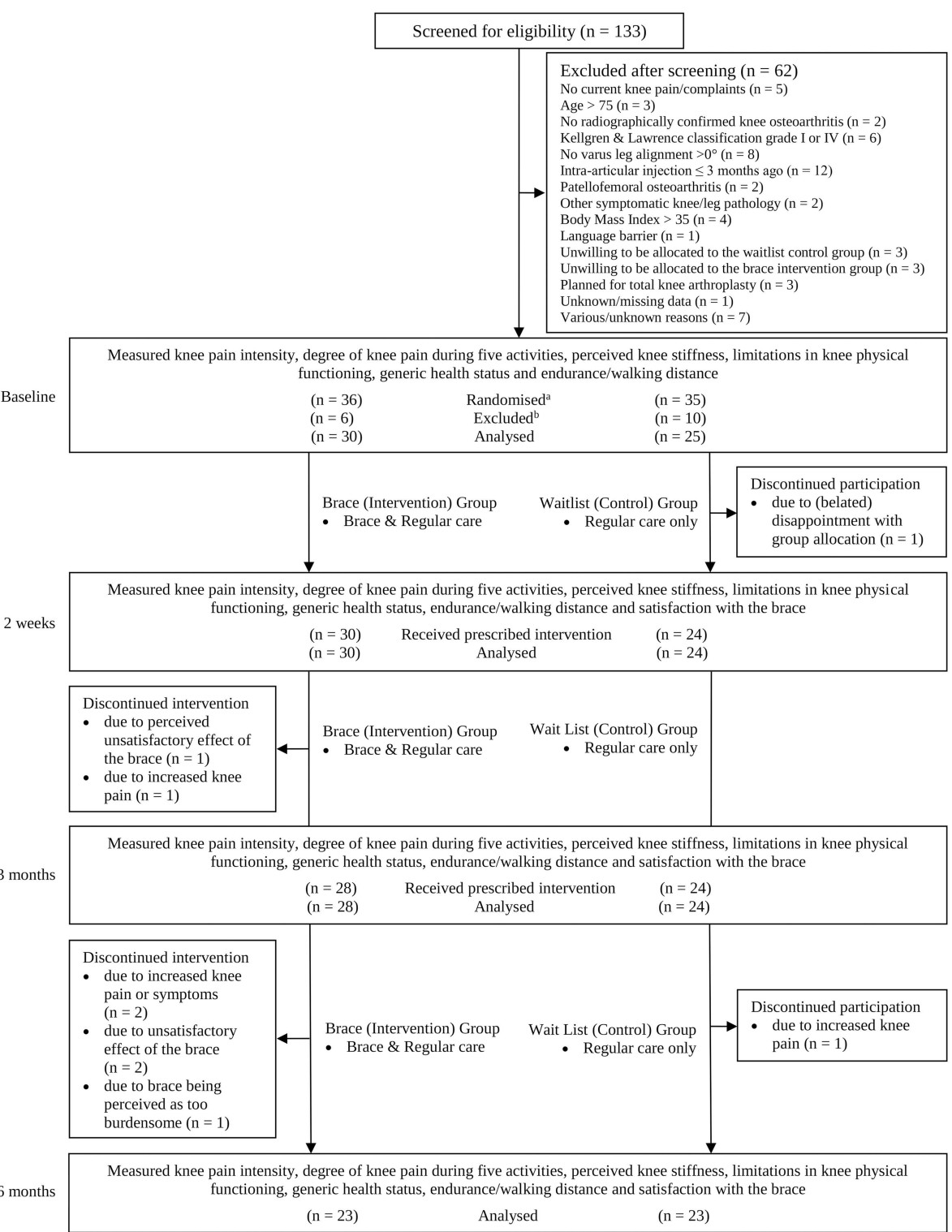

**Fig 1. Design and flow of participants through the trial.** Legend: [a] Participants were randomised prior to their baseline visit because the study's orthotist needed to be scheduled in advance and travel to the hospital to fit a brace for those allocated to the intervention group. [b] Several participants were excluded during the baseline visit, and thus after randomisation, for a variety of reasons. For example, a researcher assessed the participants' radiographs – that were taken just prior to the baseline visit – to check and measure the knee varus angle. Only during this check, it became apparent

that some participants had valgus instead of varus leg alignment (n = 4). Other screening errors included having been treated with an intra-articular injection in the knee less than three months before the baseline assessment (n = 3), a diagnosis of fibromyalgia (n = 1) or auto-immune disease (n = 1), all details that were not reported in the patients' electronic health records. Other reasons for exclusion included the refusal to be a waitlist control participant (n = 2), exceeding the cutoff BMI level of 35 sometime between screening and baseline assessment (n = 1), no more knee pain/complaints after randomisation (n = 1), no show (n = 1) and withdrawal for unclear reasons (n = 2). The 16 patients excluded for these various reasons did not undergo the baseline measurements, and this explains why there was a difference in numbers between those randomised and those ultimately analysed at baseline.

average daily brace-wearing time per week, their weekly amount of analgesics used and their weekly duration of physiotherapy exercises for the knee/leg. Participants also reported the types of physical complications they experienced and issues relating to the brace's fit during the intervention period. After every follow-up assessment, a researcher checked the participants' diaries. Subsequently, the orthotist conducted a brace check-up. The 6-month follow-up assessment concluded with a (yes/no) question asking the participants about receiving intra-articular injections during the intervention period.

### Waitlist (control) group

Participants allocated to the 6-month waitlist group were free to choose any type of regular care for their knee complaints (e.g., intra-articular injections, pain medication, physiotherapy). However, they were explicitly instructed not to use any knee brace. These participants kept diaries similar to those from the intervention group, but were not asked about brace-related issues. These diaries were also checked during every follow-up assessment. After the conclusion of the 6-month assessment, an orthotist provided each participant with the same brace as the participants from the intervention group. These participants were subsequently only followed up by the orthotist as per the usual aftercare.

### Quantitative data

Quantitative outcomes were collected at the outpatient orthopaedic department of the hospital at baseline, after two weeks and after 3 and 6 months. Both the participants and the assessor were not blinded to group allocation. The primary outcome measure was knee pain intensity [40]. This was assessed using a 10-cm VAS, completed both before ('at rest') and after performing a self-paced six-minute walk test (6-MWT) that assessed endurance by measuring the walking distance [41]. The 6-MWT has an MCID of +75 meters [42] and was performed after the participants had completed their questionnaires. At baseline, all participants performed the 6-MWT without wearing the brace, but during all subsequent assessments participants followed their allocated condition (i.e., with or without wearing the brace). The cross-validated Dutch version of the Short Form-12 (SF-12) questionnaire [43,44] asked participants about their generic health status. SF-12 sum scores were obtained by summing the scores of all items across the two (physical and emotional health) subscales. Missing values were replaced with the mean score of the subscale if ≥ 50% of the subscale items were completed. The participants' degree of knee pain during five activities, perceived knee stiffness and limitations in knee physical functioning in the past 48 hours were assessed using a validated Dutch version of the Western Ontario and McMaster Universities Osteoarthritis Index (WOMAC) questionnaire [45]. WOMAC sum scores were obtained by summing the scores of all items across the three subscales for pain, stiffness, and physical function. When one item (pain and stiffness subscales) or 1–3 items (physical function subscale) were missing, the missing value(s) were replaced with the mean of the number of available observed values for that participant. If there were ≥ 2 (pain), 2 (stiffness) or ≥ 4 (physical function) items missing, the participants' data was regarded as invalid and not used for further analysis. An effect larger than 12% of the baseline score is considered the MCID [46]. Secondary outcomes further included the participants' data regarding the self-reported use of co-interventions and brace-related data (brace intervention group only). The latter included the number of reported skin-related complications (including grazing, itching, lesions, pressure sores and redness) and issues regarding the fit

of the brace as reported in the diaries of the participants from the intervention group. The degree of satisfaction with the brace was also assessed using a 10-cm VAS.

## Qualitative data

From the participants randomised to the intervention group, a purposive sample was drawn for the post-intervention in-depth interviews to explore the participants' perceptions regarding the (use of the) brace. Sampling was based on variation in gender and adherence to the intervention. Some participants who dropped out before the 6-month assessment were purposely selected for the interview to ensure that their potentially negative perspectives were also captured for reasons of data adequacy [47,48]. A limited sample, comprising about one-third of participants from the intervention group, was considered sufficient to reach data saturation in light of the clear and relatively narrow nature and scope of the study [49], but also because the qualitative data supplemented the quantitative outcomes.

One-to-one interviews were conducted by an experienced qualitative researcher after the participants' final assessment. A mix of semi-structured open-ended and closed-ended interview questions were primarily structured around key dimensions of the Quebec User Evaluation of Satisfaction with assistive Technology (QUEST), a valid and reliable satisfaction measure assessing assistive technology outcome from the user's perspective [50–52]. Questions were formulated around aspects of a knee brace deemed most relevant in influencing user satisfaction such as comfort, safety and simplicity of use. Additional questions were structured around themes that emerged during previous work aimed at evaluating patients' perceptions regarding the use of assistive technologies. These included the themes of activities [53], aesthetics [54], (dis)comfort/ergonomics [54], physical effects and usability [55]. The interview topics are presented in Table 1. All interviews were recorded, fully transcribed verbatim by an independent commercial vendor and anonymised. Transcripts were not returned to participants for comments or corrections.

## Quantitative data collection and analysis

Patient characteristics were analysed descriptively in terms of means (standard deviation) or frequencies (percentages).

All normally distributed continuous outcomes (skewness between −1 and 1) were analysed using a per protocol, multilevel linear regression model with the 'lmer' function in the statistical programming language R to account for both within-subject and between-subject variability. Fixed effects included time (baseline, 2 weeks, 3 and 6 months) and the group (no brace/brace)-by-time interaction. Level 1 comprised all available repeated outcome measurements (full information maximum likelihood), while level 2 represented individual participants as the highest level of the model. A random intercept was assumed to account for between-participant variance at level 2. Model assumptions – normality, homogeneity of variance, and linearity of residuals at level 1, as well as normality and homogeneity of random effects at level 2, were assessed using the 'performance' and 'DHARMa' packages in R, respectively. If model assumptions were violated, the coefficient standard errors and significance levels from a robust model (fitted using the 'rlmer' function from the 'robustlmm' package) were compared to those from the original model. Subsequently, a model containing a random slope was compared to the random intercept model using a likelihood ratio test (LRT) via the 'ANOVA' function in R. This tested the null hypothesis that the simpler model provided sufficient fit. The results of the simplest adequate model were reported. Statistical significance was defined as $p < .05$ and results were visualized using figures. All model outcomes were presented alongside the individual participant data in a supporting information file.

The self-reported co-interventions were analysed descriptively in terms of means (standard deviation) or frequencies (percentages) and the between-group differences in terms of mean or percentage risk difference and their 95% confidence intervals. The weekly number of participants reporting skin-related complications and issues relating to the fit of the brace were reported in terms of frequencies (percentages).

 

**Table 1. Participant interview topics.**

| Introductory question |
| --- |
| ● What prior hopes and expectations did you have about (using) the brace? |
| **Questions about the effects of the knee brace on body functions and activities[a]** |
| ● How did the brace influence your knee complaints? Has it also influenced your knee pain/ stability/ muscle power/ mobility? |
| ● Which daily activities – that were giving you the most knee complaints – were key in your motivation to use the brace? |
| ● During which activities has the brace benefitted you most? What went well and what did not go well? |
| ● What differences, if any, did you notice between performing activities with and without the brace? |
| **Questions about the usability of the knee brace** |
| ● Can you tell me something about how easy or difficult it was to learn to use the brace? Did you run into any problems? |
| ● What are your experiences with donning and doffing the brace? |
| ● How satisfied are you with the adjustment options of the brace? Which ones have you used? How did this go? |
| ● Can you tell me something about how easy or difficult it was to use the brace (overall)? |
| **Questions about the comfort/discomfort of the knee brace** |
| ● Can you tell me something about the fit of the brace? What fittted well and what did not? |
| ● Have you had any (physical) complications from using the brace? If so, what were these? |
| ● Have you had any skin lesions, pressure sores, swelling, grazing, itching? If so, what did you do about it? |
| ● How satisfied were you with the weight of the brace? |
| **Questions about the safety of, and confidence in, the knee brace** |
| ● What are your thoughts about the structural strength/ construction of the brace? |
| ● Have you ever had any safety concerns about the brace? If so, what were these? How did this make you feel? |
| ● Have you had any brace material breakages or damages? If so, what went wrong? How has this influenced your confidence in using the brace? |
| **Questions about the aesthetics of the knee brace** |
| ● What did you think when you first saw the brace? |
| ● (How) has wearing the brace changed the way you see yourself? Has it prompted feelings of shame? Has it influenced your brace wearing time? |
| **Concluding questions about the overall experiences and satisfaction with the brace** |
| ● How would you describe your satisfaction with the brace overall? Which aspects have contributed most to your satisfaction? |
| ● Do you have any suggestions for the brace manufacturer that could improve the brace? |
| ● Will you continue to use your brace in the future? |
| ● Looking back, did the brace meet your initial expectations? |

[a]Body functions and activities terminology is derived from the World Health Organization's International Classification of Functioning, Disability and Health (ICF), version 2025−01.

## Qualitative data collection and analysis

The interview transcripts were analysed using thematic analysis taking a deductive approach [33]. This approach was chosen because the analysis was guided by existing theoretical frameworks and themes that emerged during previous research, allowing for a focused exploration of specific themes. First, two researchers independently read through the anonymised transcripts to familiarise themselves with the data. Subsequently, they collaboratively organised codes under the main (candidate) themes and subthemes by employing a stepwise categorisation process. Coding was also based on whether participants had positive or neutral and negative experiences with the brace, especially in terms of the participants' perspectives on user satisfaction and the perceived influences on body functions, activities and participation. The latter were additionally structured using the main categories and (sub)codes of the International Classification of Human Functioning, Disability and Health (ICF) [56]. The ICF is a framework that acts as a universal standard for organising information about health and related states. We deemed its use valuable for obtaining a more comprehensive understanding of the influence of the brace on health and functioning and for fostering future collaboration among different stakeholders

such as brace users, professionals prescribing braces, orthotists and brace manufacturers. Agreement on the final themes and subthemes was sought from all authors. Representative verbatim quotes that emerged during the interviews were selected to provide evidence for the formation of themes and subthemes [57]. Quotes were labelled to indicate the participants' gender (F, M), age (yr), degree of knee osteoarthritis on the Kellgren & Lawrence classification (KL 2 or 3) and the number of weeks of brace use. Only after a separate analysis, the qualitative findings were compared to the quantitative results and brought together in the interpretation phase.

## Results

### Participant flow and characteristics

Fig 1 shows the flow of participants through the trial. The first participant was included on the 7th of March 2019. In total, 133 patients were identified, screened and approached for study participation. Of these, several declined to participate because they did not like their chance of being allocated to either of the two study groups. For logistical reasons, potential participants were randomised before their baseline visit. However, because of this order of study activities another 16 patients had to be excluded after randomisation (reasons explained in the legend of Fig 1). Ultimately, 55 participants started with their allocated treatment after the baseline assessment. The final assessment of the last participant took place on the 23rd of August 2023. Eleven (37%) of the 30 participants from the brace (intervention) group were interviewed after their 6-month assessment. All the participants' baseline characteristics are presented in Table 2. There were no notable

**Table 2. The participants' baseline characteristics.**

| Characteristic | Brace (intervention) group | Waitlist (control) group | Interviewees[a] |
|---|---|---|---|
| | (*n* = 30) | (*n* = 25) | (*n* = 11) |
| Age *(yr)*, mean (SD) | 58.9 (7) | 59 (6.4) | 62.2 (9.1) |
| Gender, female | 14 (47) | 12 (48) | 5 (46) |
| BMI *(kg/m²)*, mean (SD) | 28.9 (3.1) | 28.3 (4.6) | 29 (2.4) |
| Kellgren & Lawrence classification[b] | | | |
| grade 2 | 18 (60) | 14 (56) | 6 (54.5) |
| grade 3 | 12 (40) | 11 (44) | 5 (45.5) |
| Knee varus angle *(degrees)*, mean (SD) | 5 (3.4)[c] | 4.8 (2.3) | 5.7 (2.1) |
| Analgesics use, yes | 12 (40) | 14 (56) | 5 (45.5) |
| Analgesics use *(average daily amount)*, mean (SD) | 1.4 (3.1) | 2.2 (2.7) | 1.4 (2) |
| Physiotherapy use, yes | 6 (20) | 4 (16) | 3 (27) |
| Comorbidities, yes | | | |
| Diabetes Mellitus | 2 (7) | 1 (4)[c] | 2 (18) |
| Peripheral vascular disease | 1 (3) | 3 (12)[c] | 1 (9) |
| COPD stage 3 (severe) or 4 (very severe) | 1 (3) | 1 (4)[c] | 0 (0) |
| Cardiac Decompensation | 1 (3) | 0 (0)[d] | 0 (0) |
| Rheumatoid Arthritis | 1 (3) | 1 (4)[c] | 0 (0) |

All data are frequencies (percentages) unless stated otherwise.

BMI, body mass index; COPD, chronic obstructive pulmonary disease.

[a]All interviewees were from the brace (intervention) group.

[b]The Kellgren & Lawrence classification system grades the severity of knee osteoarthritis based on radiographic findings. Only patients with grades 2 (*"Definite osteophytes and possible joint space narrowing"*) and 3 (*"Moderate multiple osteophytes, definite joint space narrowing, some sclerosis (increased bone density), and possible deformity of the bone contours"*) were eligible for study participation.

[c]Data from one participant was missing due to incomplete assessments by the assessor.

[d]Data from three participants were missing due to incomplete assessments by the assessor.

differences in characteristics between the two main groups. There were also no indications that the characteristics of the subsample of interviewees were markedly different from those of the parent sample.

## Quantitative outcomes

The results of the primary and secondary outcomes are presented in Fig 2 A-E. All individual participant data are collated in the S1 File which can be found here: https://figshare.com/s/50cacee63c01e2c2f8a6. The participants allocated to the intervention group who correctly reported their brace-wearing time (*n* = 25) wore their braces for an average of 6.6 (SD 0.48) hours per day each week over 6 months.

## Primary outcome

The multilevel linear regression analysis revealed no differences between the groups over time in knee pain at rest (Fig 2A). However, it revealed a statistically significant difference between the groups in knee pain intensity after the

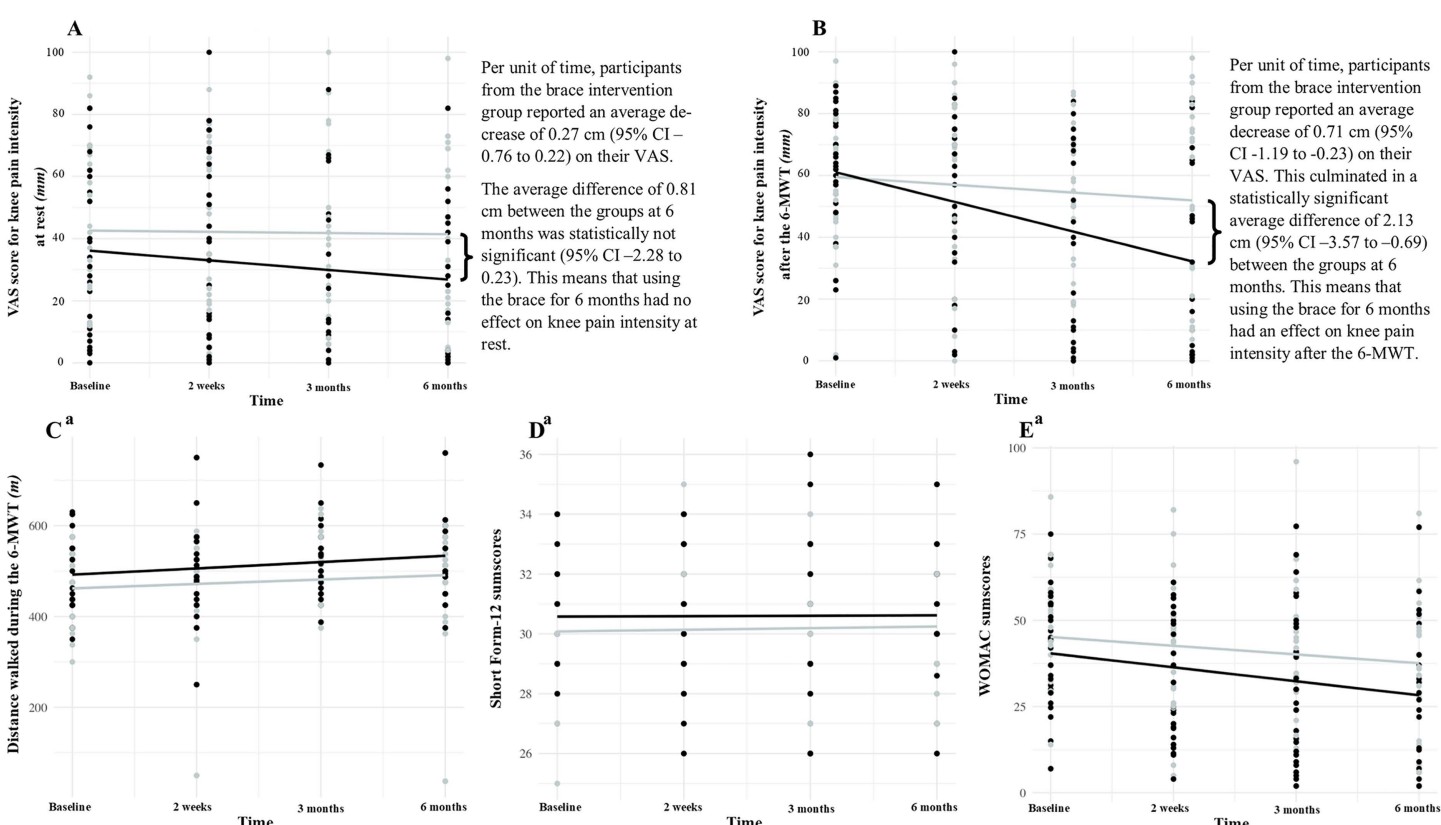

**Fig 2. A-E. The participants' VAS pain scores, walking distance covered during the 6-Minute Walk Test, Short Form-12 sum scores and Western Ontario and McMaster Universities Osteoarthritis Index (WOMAC) sum scores at baseline, 2 weeks, 3 months and 6 months.** Black dots (individual data points) and lines (group average) are data from the participants of the brace intervention group. Similarly, light grey dots and lines are data from the participants of the waitlist control group. VAS pain scores range from 0 to 10 cm where 0 means no pain and 10 means the worst pain imaginable. Short Form-12 scores range from 12 to 47 where higher values indicate a better generic health status. WOMAC sum scores range from 0 to 96 where lower scores indicate overall less knee pain during five activities, less perceived knee stiffness and less limitations in knee physical functioning. [a] Over time, participants from both groups showed a 28.9 (95% CI 4.4 to 53.3) increase in the distance covered during the 6-MWT, a negligible 0.06 change in Short Form-12 sum scores (95% CI −0.25 to 0.36) and a 7.6 (95% CI −13.8 to −1.35) decrease in WOMAC sum scores. 6-MWT, 6-Minute Walk Test; CI, confidence interval; VAS, Visual Analogue Scale; WOMAC, Western Ontario and McMaster Universities Osteoarthritis Index.

6-MWT. Participants in the brace intervention group reported an average decrease of 0.71 cm on the VAS scale for each assessment point compared to baseline (95% CI −1.19 to −0.23), culminating in an average difference of 2.13 (95% CI −3.57 to −0.69) cm on the VAS scale between the groups after 6 months (Fig 2B). The residuals for the random effect of pain at rest deviated from normality, and the assumption of homogeneity of variance was violated for pain intensity after the 6-MWT. Robust models were therefore fitted for comparison, but the results did not differ meaningfully from those of the primary analysis. The main model outcomes can be found on tab 2 of the S1 File: https://figshare.com/s/50cacee63c01e2c2f8a6

### Secondary outcomes

No differences between the groups over time were found regarding the distance covered during the 6-MWT and on the SF-12 and WOMAC (Fig 2C, 2D and 2E, respectively). After using the brace for 6 months, participants from the intervention group reported higher satisfaction scores on the VAS (mean 6.6, SD 3.2) compared to their scores at 2 weeks (mean 5.6, SD 3.1), but this increase over time was not statistically significant ($p = 0.54$, 95% CI −3.1 to 1.2).

During the first, second and third week of use, 16 (53%), 11 (37%) and 5 (17%) of the 30 brace users reported one or more skin-related complications, respectively. In the subsequent weeks, never more than three participants reported any skin-related complications per week. There were 18 (60%) and 12 (40%) participants reporting issues relating to the brace's fit in the first and second week of use, respectively. In the subsequent weeks, there were only a few participants reporting occasional issues such as the brace being too tight or slipping down during the day or the brace needing small adjustments.

### Co-interventions

Table 3 shows that a comparable number of participants from both groups used analgesics during the study, but that those from the brace intervention group ingested significantly lower amounts (mean difference −33.2, 95% CI −59.3 to −7.1). To assess potential confounding, the amount of ingested analgesics was added to the models for the primary and secondary outcomes. However, the adjusted models did not outperform the simpler model that included only a random intercept. There were no statistically significant differences between the groups in any of the other co-interventions. A post-study review of participants' electronic health records in February 2024 further revealed that, of all analysable participants with Kellgren & Lawrence classification grade 3, seven of 12 (58%) from the intervention group and three of 11 (27%) from the control group had received TKA or UKA surgery. It is also noteworthy that all interviewees with

**Table 3. The participants' data regarding the self-reported use of co-interventions during the 6-month study.**

| Outcome | Groups | | Difference between groups (95% CI) |
|---|---|---|---|
| | Brace (intervention) group | Wait list (control) group | |
| Use of analgesics, yes | 20 (83)[a] | 17 (81)[b] | RD 2% (−20% to 25%) |
| Analgesics used *(total amount ingested)*, mean (SD) | 20.8 (39.3)[a] | 54 (47.6)[b] | MD −33.2 (−59.3 to −7.1) |
| Intra-articular injections received, yes | 1 (7)[c] | 1 (7)[c] | RD 0% (−20% to 20%) |
| Use of PT exercises for the knee/leg, yes | 6 (25)[a] | 8 (47)[d] | RD 22% (−7% to 50%) |
| PT exercises for the knee/leg per *(total number of minutes)*, mean (SD) | 542.5 (422.7)[e] | 451.3 (471.5)[f] | MD 91.3 (−440.4 to 622.9) |

All data are frequencies (percentages) unless stated otherwise.

95% CI, 95% confidence interval; MD, mean difference; PT, physiotherapy; RD, percentage risk difference.

[a]$n = 24$, [b]$n = 21$, [c]$n = 15$, [d]$n = 17$, [e]$n = 6$, [f]$n = 8$.

grade 3 underwent surgery after the study. In comparison, only two of 18 (11%) and two of 14 (14%) of all analysable participants from the intervention and control group with Kellgren & Lawrence classification grade 2 had received knee surgery, respectively.

### Qualitative findings

During the study, a researcher experienced in qualitative research (LDdJ) continuously analysed the interview data as it was being collected and confirmed that no new information emerged after about 10 interviews. An overview of all themes, subthemes and the original verbatim Dutch and their English clean verbatim translations of the quotes emerging from the interviews can be found in S1 Table (https://figshare.com/s/20212f8bb7fba462d17f).

### Prior hopes and expectations about (using) the brace

The majority of participants said that they had hoped and expected (S1 Table, Table 1A) that using the brace would decrease their knee pain, with some hoping that it would also enable them to perform activities such as walking or playing sports as they did before the knee complaints:

*I had the expectation that it would maybe ease the pain a little. I was very hopeful that I would be selected for a brace because I thought 'this is it!' I thought 'then I can walk and do things in and about the house.'* (F, 67, KL 3, 24 weeks of brace use)

A few participants expected better knee stability/support or that the brace would assist in keeping their leg axis misalignment in check. Some participants expressed quite specific or general hopes and expectations, such as that the brace would prevent locking of the knee or that the knee complaints would disappear altogether. One participant hoped that knee surgery could be prevented and that it would make his life easier:

*Well, the expectations were that I would be better, that I would find support from it. And that I wouldn't need to undergo surgery because of it. I had expected that I would actually be able to do more. My expectations may have been quite high, yeah, that wearing it would make my life a bit easier.* (M, 52, KL 3, unknown number of weeks of brace use)

Two participants said they did not have any prior expectations.

### Aesthetics

The participants' first impressions of the brace were captured under the main deductive theme of aesthetics (S1 Table, Table 1B). The first thought crossing most participants' minds when seeing the brace for the first time, was that it was bigger than they expected:

*When I saw it, oh yes, when the orthotist first brought it in, I did think 'wow, the size of that thing!' But that feeling quickly faded.* (F, 74, KL 2, 22 weeks of intermittent brace use)

### Perceived influence on body functions, activities and participation

One theme described the participants perceived changes in body functions (S1 Table, Table 1C). Under this theme, the subthemes (and ICF subcodes) identified were pain, stability, mobility, muscle power, control of voluntary movements, confidence and functions of lymphatic vessels.

The identified subtheme of pain was describing the highest number of opposing perspectives. While about half of the interviewed participants perceived positive changes such as a decrease in pain, a few others perceived no or only short-lived decreases in pain or even an increase in pain:

*It doesn't take the pain away but compared to the time when I didn't have the brace, the pain is less. Much less. The pain during stair climbing has improved significantly.* (M, 75, KL 2, 23 weeks of brace use)

*I work in healthcare, so I walk a lot, about 8 hours, 9 hours a day. At first I thought 'oh, it helps.' But no, I still feel like I didn't notice any difference. Despite wearing the brace, I still had pain.* (F, 59, KL 3, 8 weeks of brace use)

Two other subthemes identified under the body functions theme were mobility and muscle power. The few participants commenting on this said that the brace had not or only somewhat restricted their overall knee movements. In the same vein, wearing the brace had also not negatively influenced muscle power. However, two participants said that wearing the brace led to some signs of leg muscle atrophy:

*I went there and the sports therapist said 'those muscles, you can tell they're getting thinner.' So, that brace does absorb a lot, but those muscles are becoming less active.* (M, 51, KL 2, 1 week of brace use)

For the majority of participants using the brace was perceived to have had a positive influence on their knee stability and control over some specific voluntary movements:

*I did have a bit more stability. Some support. Support for my leg, for my knee.* (F, 59, KL 3, 8 weeks of brace use)

*With the brace it's easier to make sharp turns with the leg because the brace is sturdy and it doesn't allow for rotation in the knee. Then you also have less pain.* (M, 56, KL 2, 24 weeks of brace use)

One participant reported an increased confidence in their knee and another reported that that their knee was less swollen when wearing the brace.

Another theme described the participants perceived influence of the brace on their activities. Under this theme, the identified subthemes (and ICF subcodes) were activities related to walking, lifting and carrying objects, kneeling and moving around using transportation.

Except for two participants, who were neutral in their experiences, several others felt that using the brace had positively influenced their walking activities. These included walking up and down stairs for some. The brace also supported some participants in lifting and carrying objects, using a wheelbarrow or stepping over a garden fence during their gardening activities:

*Because I could walk a bit better with that brace. I used to enjoy walking a lot, but I haven't been doing that for a while now. And I always used it when working in the garden, that went quite well. Then I had more stability, for example when I was walking with the wheelbarrow, that went well. Better. So, when I was doing some heavier activities it was easier with the brace than without it.* (M, 52, KL 3, unknown number of weeks of brace use)

For one participant the brace was supporting the leg movement during cycling, but a few other participants complained that the brace was preventing them from kneeling and a hindrance when trying to get into the car, getting on and off a forklift or during cycling:

*I operate a forklift and things like that. When getting on and off the forklift you can't position the leg like you normally do. That's a hindrance, that's a pity.* (M, 60, KL 3, 24 weeks of brace use)

With regards to the theme of perceived influence of the brace on participation, only two participants specifically commented on how using the brace either positively or negatively influenced some of their work-related activities. As such, these comments were identified under the subtheme (and ICF subcodes) of remunerative employment.

**User satisfaction**

An overview of the participants' perspectives relating to the main theme of user satisfaction with the brace is provided in S1 Table, Table 1D. Under this theme, the deductive subthemes identified were simplicity of use, dimensions (i.e., fit), comfort in terms of physical complications and in terms of psychological well-being, adjustments, durability, weight and safety.

The identified subtheme of simplicity of use elicited a few opposing perspectives. Most participants indicated they had not had any or only minimal problems in using, learning to use and donning and doffing the brace. However, despite repeated instructions from the orthotist, a few participants kept struggling with donning and doffing the brace:

*I find putting the brace on and taking it off quite difficult before you get this feeling of it fitting properly. At first the proper position was neatly marked on my leg. But even then it's still difficult to get it in exactly the right position. I find that really difficult. You do feel that at some point, but it takes time before you get it right. I found getting in and out of all those straps also difficult. The convenience of handling is limited. I just find it very inconvenient to use.* (F, 75, KL 3, 4 weeks of intermittent brace use)

The subtheme of dimensions captured the participants' perspectives about the brace's fit in terms of convenience of its height, width and length. Although some participants reported to have had some minor issues such as the brace sometimes slipping down, the majority of participants said the brace had a good fit:

*Yeah, the brace fits me well. Like I said, sometimes I have to remember to adjust it a bit and slide it up. But other than that, it fits well.* (F, 55, KL 2, 24 weeks of brace use)

The participants' perceptions about the brace's comfort were of a more complex nature. Almost all participants experienced one or more physical complications such as minor wounds, pressure sores, redness of the skin or pain. However, the comments of many of these participants suggested that they were not overly bothered by these complications because of their transient nature:

*Well, what I do have is, at the end of the day there's a little indentation at the back of my leg. But that will clear overnight.* (F, 55, KL 2, 24 weeks of brace use)

A few participants asserted that the brace should be worn over the pants instead of on bare skin after having experienced physical complications such as sweating or other irritations of the skin:

*I also wore the brace under my pants for a while. Then I got a bit of a red rash on my skin. I didn't have that then wearing the brace over my pants.* (M, 56, KL 2, 24 weeks of brace use)

While reflecting on the brace's comfort in terms of psychological well-being, most participants comments suggested that wearing the brace did not lead to feelings of shame:

*Never, not one minute have I felt ashamed of the brace. No, wearing the brace hasn't affected the image I have of myself, you get used to it very quickly. People look or ask questions about it. And then I think 'yeah, it's just part of me.'* (F, 67, KL 3, 24 weeks of brace use)

In one case, a participant's spouse who was present during the interview expressed some initial feelings of second-hand embarrassment, which turned into more positive feelings over time:

*I felt a bit ashamed. It just looks ridiculous wearing them on top of the pants like that. Just wear it underneath! But I was soon cured of that feeling when her physical complications came to light. Now I don't feel any embarrassment at all anymore when she's walking around with that thing. In fact, sometimes I think it is kind of cool…*(Spouse of F, 74, KL 2, 22 weeks of intermittent brace use)

Except for one, all participants were positive about options to set and adjust the different components of the brace such as the Velcro straps and the varus/valgus setting by means of an Allen key:

*In terms of adjustment options, it was good. With those Velcro straps you open it up and then you loosen the brace. If it's too loose, you just strap it in a bit more. If you have the Allen key with you, you can tighten it more easily.* (M, 56, KL 2, 24 weeks of brace use)

Apart from some minor issues with quick-fit fasteners sometimes coming loose, most participants felt positive about the brace's durability. This subtheme included perceptions about how robust and sturdy the brace was or if participants had experienced any breakages and damages:

*You occasionally get caught behind that clip on the side, causing the brace to come loose.* (F, 67, KL 3, 24 weeks of brace use)

*I find the stability of the brace fantastic. It provides stability all around. No, the brace has never been broken or damaged during use.* (M, 75, KL 2, 23 weeks of brace use)

Apart from two participants, who felt that the brace was heavy, weight was not considered an issue of concern for all others. In response to a closed-ended question about the safety of the brace, none of the participants expressed any concerns in terms of buckling through the brace for whatever reason.

### Overall experiences and satisfaction with the brace

Participants responses to the concluding interview questions (S1 Table, Table 1E) elicited a variety of remarks that echoed the participants' sometimes negative, mixed or positive overall experiences and satisfaction with the brace:

*I've used the brace for about a month and a half. I found that to be too much. That's a bit demotivating. When too many things work against you, you indeed think 'yeah, I'm done with it.' So, I'm not that satisfied actually. The disadvantages were greater than the benefits. You don't notice the benefits in the short term, but you immediately notice the disadvantages. My dissatisfaction is mostly influenced by the fact that it wasn't easy to put the brace on myself. No. I hope it won't be necessary to wear the brace in the future. I'm having a total knee replacement and hopefully that will improve things.* (F, 75, KL 3, 4 weeks of intermittent brace use)

*I think the brace is a very good invention, but I haven't actually had the benefit that I had hoped for. I think I will keep wearing it as long as there is no other solution.* (F, 67, KL 3, 24 weeks of brace use)

*Yes, my expectations were met. As soon as I have the brace on I have less pain. It's sturdy. I find it comfortable to walk with. It's like a straightjacket for the knee. When it's on I can feel the benefits. If I don't wear it, I'm just shooting myself in the foot. It has helped, but it hasn't solved the problem. Yes, I will continue wearing the brace in the time to come. I don't know when I should see the orthopaedic surgeon again to see what needs to be done next. If he says 'I'll put in a*

*new knee' then I'll just keep the brace because maybe I can save it for my other knee.* (M, 60, KL 2, 24 weeks of brace use)

Most participants were more or less sure they would keep using the brace in the future. Some other participants were sceptical about whether the brace had sufficiently benefitted them. Only two participants said they were convinced they would never wear the brace again after the study. Overall, some of the findings also suggested that the participants with a higher degree of knee OA (i.e., Kellgren & Lawrence classification grade 3) were less satisfied with the effects of the brace on their knee pain and were perceiving more activity limitations in terms of kneeling or moving around using transportation.

Finally, some participants felt that the brace could be improved by incorporating some kind of aid or visual reference point that would allow them to consistently put the brace in the exact same position. Some participants also felt that the (grips of the) quick-fit fasteners could be improved because sometimes these came loose, e.g., when getting stuck behind a chair or clothing. One participant suggested to replace the quick-fit fasteners with something akin to a dial-fit closure adjustment system. Minor other improvements suggested by some individual participants were to issue the brace with a clearer paper manual, offering more neutral colors and using even lighter materials. One participant, who was 160 cm tall and the second-shortest of the entire sample, suggested to make the lowest fastener and Velcro-strap easier to reach for short people.

### Triangulation of quantitative and qualitative data

A comprehensive triangulation of quantitative outcomes and qualitative (sub)themes in overlapping areas of data is provided in S2 Table (https://figshare.com/s/7c3894a86e4b00872d41). In summary, quantitative data showed limited group-level differences in general pain, physical function, or health status. Brace users only reported significantly less pain after the 6-MWT and used fewer analgesics. In contrast, findings varied across participants with many participants qualitatively describing improvements in pain, knee stability and movement control not captured by quantitative outcomes. Several participants also reported functional gains in daily activities, despite no measurable change in 6-MWT or WOMAC scores. Findings converged in suggesting that participants with more severe OA perceived the brace as less effective, which corresponded with higher arthroplasty rates in this subgroup. Findings were consistent across participants in terms of skin-related complications and fit issues, with most being transient and not significantly impacting satisfaction. Satisfaction with the brace increased meaningfully, but the change was not statistically significant and not reflected in VAS scores.

### Discussion

In this study, the effectiveness of a valgus brace was evaluated by triangulating the results of objective measures of knee pain, knee functioning and subjective perspectives of the participants with medial compartment knee OA who used the brace. The quantitative results of this study strongly suggest that the brace was effective in decreasing knee pain intensity during walking activities. There were no further statistically significant differences between the brace users and those who were on a waitlist for a brace in terms of knee pain at rest and knee physical functioning. It is worth noting that the participants of the brace intervention group used less analgesics during the study, and that the higher usage of analgesics by participants in the waitlist control group may have influenced their VAS scores for pain. Overall, the study had insufficient power to detect differences in the key outcomes between groups, increasing the chance of a type II error. Nevertheless, the qualitative findings showed that about half of the interviewed participants believed that using the brace positively impacted several body functions (including pain) and activities. Given the discrepancy between the quantitative results and the qualitative findings, some important issues need to be discussed.

### Comparison of the quantitative and qualitative data regarding body functions and activities

The results of systematic reviews have shown that valgus braces can have positive effects for some patients with medial compartment OA in terms of knee pain, knee function, activity levels and quality of life [18,25,58,59]. Compared to these

results, the quantitative results of our study only confirmed that 6 months of brace use led to significantly less knee pain when assessed directly after a functional walking test. This was a robust finding given that the statistically significant difference between the groups was found with a sample size smaller than calculated a priori. The 21 mm decrease in VAS scores from the baseline to the 6-month assessment also exceeded the 15 mm minimally clinically important difference reported in the literature [37,38]. This additionally suggests that the decrease in pain was relevant for our participants. Other than this result, there were no statistically significant differences between the groups in any of the other outcomes. However, the study's qualitative findings painted a different picture regarding the latter. Although about half of the interviewed participants felt that using the brace had little to no or even negative effects on their knee pain in general, the other half experienced pain relief from using it. In addition, a majority of participants also perceived positive influences on their knee stability, control over some specific voluntary movements and the various walking activities. These latter findings were not corroborated by the VAS scores for pain at rest and the results on the WOMAC, but it is worth noting that over time these scores diverged between the groups. With a longer treatment time or with a larger sample size (and a smaller 95% CI), a statistically significant difference between the groups might have emerged. As recommended [60,61], other possible reasons for discrepancies between the quantitative results and the qualitative findings were also explored.

The VAS quantified knee pain intensity and the WOMAC primarily quantified knee pain during rest, movement and various activities. Interview questions about the impact of the knee brace on body functions and activities strongly related to both these outcomes. This suggests that any discrepancies between the quantitative and qualitative findings are not due to having assessed different phenomena. There was also nothing to suggest that the discrepancies were due to biased sampling, as the participants' baseline characteristics indicated that the interviewees were representative of the entire sample. Having ruled out these reasons for inter-method discrepancies, there is a chance that averaging the VAS pain intensity and WOMAC data across all brace users has obscured individual variations in treatment response. This suggests that these outcomes tools may lack sensitivity to capture subjective improvements valued by patients. In addition, results of previous studies have suggested that a higher degree of knee OA [62,63] could negatively impact or potentially confound the perceived effectiveness of a knee brace. Some of our qualitative data may also have indicated that the brace was less effective in terms of knee pain and activity limitations for the participants with a higher degree of knee OA. This finding was further supported by the post-study observation that a larger proportion of brace users with Kellgren & Lawrence classification grade 3 underwent knee arthroplasty surgery at some point after the study. Ideally, a sensitivity analysis was performed to analyse if the participants' degree of knee OA indeed confounded the results, but we purposely refrained from doing so because the resulting sample sizes would be too small to draw any meaningful conclusions.

In terms of influences on body functions and activities, the overall results of this study suggest that, while some participants benefited from using the valgus brace by experiencing reduced knee pain and improved function, others did not. A wider range of patient experiences was uncovered by applying a mixed methodology, experiences which might not have been captured if the study had used a more 'traditional' approach of using quantitative measures only. Relying solely on quantitative data would have led to the conclusion that the valgus brace had limited effectiveness. This study once more exemplifies how integrating both quantitative and qualitative data results in conclusions diverging from those drawn solely from either method [60] and underscores the need to integrate qualitative insights when interpreting trial outcomes, especially for interventions with variable individual responses.

## User satisfaction

Satisfaction of people using assistive technology devices, of which knee braces are just one example, depends on many factors [50]. For example, despite improvements in gait and balance, many patients were not using their prescribed assistive devices because they were dissatisfied with its properties, aesthetics or fit. Barriers to wearing the devices included perceived issues with donning and doffing, their weight and perceived interference with daily activities [64]. This may explain why the compliance of using assistive devices in general [64], and offloader braces in particular, is low [65,66]. By

exploring the user experiences of a subsample of knee brace users during this RCT, we not only collected in-depth information that supplemented the quantitative data, but also collected information that could uncover the participants' reasons for use, non-use, satisfaction and dissatisfaction.

The majority of interviewed participants reported that wearing the brace did not adversely affect their self-image or induce feelings of shame. Additionally, most participants expressed satisfaction with the available options to self-adjust the brace and its lightweight design. While some participants had occasional issues with quick-fit fasteners coming loose, no braces were damaged and no patients experienced any buckling while wearing the brace. These latter experiences, as well as the overall positive perceptions regarding the brace's structural strength, may explain why none of the participants expressed any concerns about the brace's safety. Aspects of the brace that elicited some negative perspectives, and which probably contributed to feelings of dissatisfaction for some, were related to problems with using and learning to use the brace. More specifically, some were having difficulties putting the brace on correctly. Comparable difficulties were previously reported by other unloader brace users [66,67]. Addressing this issue was mooted by our participants as necessary to enhance the brace's simplicity of use. As was also observed during previous studies [62,66–69], almost all interviewed participants reported minor transient skin problems. Arguably, such complications can be caused by a poor or suboptimal fit of a brace [66,69]. However, because only a few negative perspectives emerged about the brace's fit, the skin-related complications were most likely caused by increased friction and pressure on the skin associated with the breaking-in period. This assumption is supported by the quantitative results about most of the physical complications occurring in the first three weeks of brace use. Overall, the majority of participants seemed quite happy with most of those aspects of the brace that were deemed most relevant in influencing the degree of user satisfaction. This finding was also reflected by an increase in VAS scores for satisfaction with the brace from the 2-week to the 6-month assessment. However, although the 10 mm change in VAS scores over time was within the 7–11 mm range considered to be clinically relevant [70], this change was not statistically significant.

## Strengths, limitations and future research

To our knowledge, our randomised controlled trial is the first in its kind to investigate the effectiveness of a valgus brace on knee pain and activity limitations in patients with medial compartment knee OA, while at the same time qualitatively exploring some brace users' perceptions regarding the use of their brace. Although several previous studies have reported on brace users' experiences and satisfaction, this information was mainly quantitative in nature. Using a combination of quantitative and qualitative factors has been deemed essential in driving advancements in knee brace design [71]. This is why we delved more deeply into the perspectives of brace users by focusing on the key dimensions of QUEST and incorporating themes that emerged during prior research.

The first main limitation of this single-centre study was that the participants and assessors were not blinded to group allocation, which introduced a risk of measurement bias especially in subjective outcomes such as the VAS pain scores. Another main limitation was that the study was underpowered. Factors contributing to the latter included the logistical challenge of the need to randomise participants before their baseline visit and the negative impact of COVID-19-related social restrictions on patient recruitment and the researchers' ability to conduct several follow-up assessments. This resulted in a smaller sample size than anticipated and some data missing completely at random (MCAR) on different measurement occasions. The study's underpowered nature may have reduced its ability to detect potentially true effects or differences in outcomes, increasing the risk of a type II error. In addition, a few eligible participants declined, or may have declined, to participate once they learned they had been randomised to the waitlist control group. This may have led to attrition bias. Because this was also noted during a previous study [27], we had hoped to avoid this by allowing participants from the waitlist control group to undergo any type of treatment for their knee complaints except a knee brace. Although this control intervention arguably mirrored real-world conditions, it may have resulted in significant variability, confounding and insufficient treatment contrast between the two study groups. All these factors potentially increased the

risk of a type II error. The study's findings may also have limited generalizability to broader patient groups or different clinical settings due to the wide age range (40–75 years), varying BMI profiles and potential differences in comorbidities among participants. Also, apart from checking the participants' diaries during each follow-up assessment, no additional safeguards were implemented to verify the accuracy of the self-reported brace-wearing time and co-interventions, leaving recall bias and social desirability bias as potential concerns.

We recommend conducting well-powered and high-quality studies, with follow-up assessments beyond 6 months of brace use, to improve evaluations of the effectiveness of individually fitted knee braces on body functions, activities and participation. Further in-depth exploration of the brace users' perspectives, including factors that affect these perspectives, may help to optimise the design and usage of valgus braces. Large-sample research is also warranted to explore if patient factors such as age, physical activity, BMI [25], body length, wearing time, perceived fit and comfort, and the degree of knee OA affect knee brace effectiveness. Observations from this study suggest that the degree of OA may influence brace effectiveness, but larger patient samples are needed to enable meaningful sensitivity analyses. The results of studies with larger samples will arguably provide a better understanding of which patients with OA benefit most from assistive technologies such as knee braces, and so these can be improved further. This may aide the different knee brace stakeholders in making informed decisions and choices regarding the design and prescription of knee braces with the ultimate aim of improving patient outcomes and quality of care.

## Conclusion

The quantitative results of this study suggest that the use of a valgus brace was effective in decreasing knee pain intensity during walking activities with the brace in patients with medial compartment knee OA. Although using the brace had no statistically significant effects on knee pain at rest and knee physical functioning, the qualitative findings of the study showed that about half of the interviewed participants believed that using the brace positively impacted several of their body functions (including pain) and activities. Some post-study observations and qualitative findings may indicate that a higher degree of knee OA negatively influenced the effectiveness and user experiences of the valgus brace.

## Supporting information

**S1 File. Supporting Information File 1: quantitative data.** Tab 1) Individual outcome data. Tab 2) Mixed model results.
(XLSX)

**S2 File. CONSORT checklist.**
(DOCX)

**S1 Table. Supporting Information Table 1A-E: qualitative data.**
(PDF)

**S2 Table. Triangulation of quantitative outcomes and qualitative (sub)themes in overlapping areas of data.**
(PDF)

## Author contributions

**Conceptualization:** Corné J.M. van Loon.

**Data curation:** Lex D. de Jong, Babette C. van der Zwaard, Matthijs Y.H. van Blommestein.

**Formal analysis:** Lex D. de Jong, Babette C. van der Zwaard, Corné J.M. van Loon.

**Funding acquisition:** Corné J.M. van Loon.

**Investigation:** Lex D. de Jong.

**Methodology:** Corné J.M. van Loon.

**Project administration:** Lex D. de Jong.

**Supervision:** Lex D. de Jong, Corné J.M. van Loon.

**Validation:** Lex D. de Jong, Matthijs Y.H. van Blommestein.

**Visualization:** Lex D. de Jong, Babette C. van der Zwaard.

**Writing – original draft:** Lex D. de Jong.

**Writing – review & editing:** Babette C. van der Zwaard, Matthijs Y.H. van Blommestein, Corné J.M. van Loon.

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
