## [Decision Letter · Decision Letter 0]

2 May 2025

Effectiveness and user experiences of a valgus brace in patients with knee osteoarthritis: A mixed-method randomised controlled trial.

PLOS ONE

Dear Dr.  de Jong,

Thank you for submitting your manuscript to PLOS ONE. After careful consideration, we feel that it has merit but does not fully meet PLOS ONE’s publication criteria as it currently stands. Therefore, we invite you to submit a revised version of the manuscript that addresses the points raised during the review process.

We look forward to receiving your revised manuscript.

Kind regards,

Taher Babaee

Academic Editor

PLOS ONE

Journal Requirements:

Reviewers' comments:

Reviewer's Responses to Questions

**Comments to the Author**

1. Is the manuscript technically sound, and do the data support the conclusions?

Reviewer #1: Partly

Reviewer #2: Yes

2. Has the statistical analysis been performed appropriately and rigorously?

Reviewer #1: No

Reviewer #2: Yes

3. Have the authors made all data underlying the findings in their manuscript fully available?

Reviewer #1: Yes

Reviewer #2: Yes

4. Is the manuscript presented in an intelligible fashion and written in standard English?

Reviewer #1: Yes

Reviewer #2: Yes

Reviewer #1: This manuscript presents a mixed method randomized controlled trial (RCT) evaluating the effectiveness of a valgus brace in patients with medial compartment knee osteoarthritis (OA). The primary outcome was knee pain intensity at six months, assessed via a 10-cm Visual Analogue Scale (VAS). Secondary outcomes included walking distance, generic health status, knee functioning, and patient satisfaction. The study also incorporated qualitative interviews to explore user perceptions of the brace. The quantitative analysis revealed a statistically significant and clinically meaningful reduction in knee pain intensity after a walk test, but no significant effects on other outcomes. The qualitative findings indicated mixed but generally positive experiences regarding the brace’s usability and effectiveness. Strengths of the study include the appropriate use of a mixed-method RCT, a robust statistical approach using multilevel linear regression, clinically relevant outcome measures, consideration of confounders, and prospective trial registration ensuring transparency. However, I have the following concerns:

1. The study was underpowered due to an insufficient sample size (n=23 per group) compared to the initial target of 80 participants, increasing the likelihood of Type II errors, and the sample size calculation does not appear to account for potential attrition rates beyond a general 15% estimate. The authors should clearly state how the final sample size impacts the interpretation of non-significant findings and conduct a post hoc power analysis to determine if the study was sufficiently powered to detect meaningful differences in secondary outcomes.

2. The manuscript does not provide a clear strategy for handling missing data beyond simple mean imputation for some questionnaire-based measures (e.g., SF-12 and WOMAC), and it is unclear whether missing data were missing completely at random (MCAR), missing at random (MAR), or missing not at random (MNAR). I would recommend using multiple imputation or sensitivity analyses to assess the impact of missing data on the robustness of results and report the extent of missing data for each outcome measure while justifying the chosen imputation method.

3. Multiple statistical tests were performed, yet no correction for multiple comparisons (e.g., Bonferroni or Benjamini-Hochberg) is mentioned, increasing the risk of Type I errors. The authors should apply an appropriate correction method for multiple comparisons or justify why it was not deemed necessary and consider reporting effect sizes alongside p-values to provide a clearer clinical interpretation.

4. While multilevel modeling is an appropriate choice, the manuscript does not provide details on why a random intercept model was preferred over alternatives such as random slopes or mixed-effects models with covariate adjustments, and it is unclear whether model assumptions (e.g., normality of residuals, homoscedasticity) were tested. I would recommend that the authors should provide diagnostic checks (e.g., residual plots, normality tests) to confirm model appropriateness and consider alternative models, such as a repeated-measures ANCOVA adjusting for baseline covariates.

5. The qualitative findings suggest benefits not detected in quantitative analysis, raising concerns about potential bias in either data collection or interpretation, and no integration framework (e.g., triangulation analysis) is used to reconcile these discrepancies. I recommend that the authors consider using a joint display table to directly compare quantitative and qualitative findings and discuss how selection bias in the qualitative sample might have influenced results.

6. Although co-interventions (e.g., analgesic use, physiotherapy) were recorded, no clear sensitivity analyses were conducted to assess their impact on primary outcomes. Can authors conduct subgroup or sensitivity analyses to explore whether co-interventions influenced the observed effects and report whether these factors were included as covariates in the regression models?

7. The study only evaluates outcomes up to six months, which may not be sufficient to assess long-term efficacy and adherence to brace use. The Authors should discuss the need for follow-up studies examining long-term effects beyond six months and consider analyzing long-term adherence and its relationship with effectiveness.

This study employs a robust mixed-method approach to evaluating the effectiveness of a valgus brace in knee OA. However, several methodological and statistical concerns, including an underpowered sample, missing data handling, confounding, and multiple comparison adjustments, need to be addressed. Implementing the recommended improvements will strengthen the validity and interpretability of the findings.

Reviewer #2: I have carefully reviewed the manuscript titled “Effectiveness and user experiences of a valgus brace in patients with knee osteoarthritis: A mixed-method randomised controlled trial.”

While this is an important and valuable contribution, several methodological, reporting, and presentation issues need to be addressed before the manuscript can be considered for publication.

Below are my detailed comments:

Major Issues

Abstract Reporting:

The effect size of the main outcome should be explicitly reported in the Abstract alongside the p-values.

The number of participants per group (n=23) should be stated in the Abstract Methods section, not in the Results.

Effect Size Interpretation in Main Text:

Beyond statistical significance, the authors should interpret and discuss the clinical magnitude of the effects (e.g., Cohen’s d or appropriate effect size) in the Discussion section.

Insufficient Description of the Brace:

Although the SecuTec® OA brace is mentioned, there is no clear orthosis and technical description of its biomechanical mechanism.

Such details are essential for replication and clinical relevance.

Reference Needed for Literature Gap:

Line 87: The claim about the lack of qualitative studies evaluating orthotic and assistive devices requires a supporting reference.

Wide Age Range Needs Acknowledgment:

Line 100–102: Including participants aged 40–75 introduces potential heterogeneity. This should be acknowledged explicitly as a limitation.

[NEW] Limited Generalizability:

The study’s findings may have limited generalizability due to the wide age range (40–75 years), varying BMI profiles (with an exclusion criterion of BMI >35), and potential differences in disease severity or comorbidities among participants. This aspect should be critically addressed as a limitation to provide a more balanced interpretation of the results.

Reference Needed for BMI Cutoff:

Line 110: A reference supporting the exclusion criterion of BMI >35 is needed to justify its clinical relevance.

Control Group Design Concerns:

Line 173–177: Allowing the control group to pursue any standard treatment (physiotherapy, injections) except bracing may introduce significant variability and confounding. This needs to be critically discussed as a methodological limitation.

Tables Formatting and Completeness:

Baseline Characteristics Table (Table 2): Should report p-values to confirm baseline comparability between groups.

Units of Measurement: All variables should have clearly stated units (e.g., meters, degrees, cm).

Footnotes: Should provide expanded explanations regarding missing data.

Consistency: Data formats (mean (SD), n (%)) should be standardized across all tables.

Adjusted Analyses: Table 3 (co-interventions) should ideally include or at least discuss adjustment analyses for potential confounders.

Figures Quality and Completeness:

Figures, especially the CONSORT diagram, should be provided at higher resolution (minimum 300 dpi) to ensure clarity.

Figure legends should be expanded to be fully self-explanatory.

In-text references to figures should be clearer.

Inclusion of a schematic illustration or technical diagram of the valgus brace’s mechanism would greatly improve readers’ understanding.

Sample Size and Power Issues:

The final analyzed sample size (n=46) was notably lower than planned, leading to an underpowered study. The risk of Type II error should be more strongly emphasized.

Randomization Before Eligibility Confirmation:

Line 124–149: Randomizing participants before full eligibility confirmation led to post-randomization exclusions, which can introduce selection bias.

Lack of Blinding:

Line 127: Neither participants nor assessors were blinded, increasing the risk of measurement bias, particularly for subjective outcomes like VAS.

The language is professional and clear, with only minor grammatical improvements needed.

Summary Recommendation

The manuscript addresses a clinically important question with a valuable mixed-methods design.

However, substantial revisions are needed to address issues regarding outcome measurement appropriateness, complete and transparent reporting (especially in tables and figures), stronger methodological justifications, and clearer acknowledgment of study limitations.

Addressing these concerns will significantly enhance the manuscript’s scientific rigor, transparency, and potential clinical impact.

Discussion Section Suggestions:

In the Discussion section, the authors should expand on the limitations identified, including the potential impact of the wide age range (40–75 years), varying BMI profiles (despite the exclusion of BMI >35), and other participant characteristics (e.g., disease severity or comorbidities). These factors may limit the generalizability of the findings to broader populations or different clinical settings. The authors should discuss how these heterogeneities might influence the applicability of the results and suggest directions for future research, such as conducting subgroup analyses or targeting more homogeneous cohorts to validate the findings.

**Do you want your identity to be public for this peer review?** For information about this choice, including consent withdrawal, please see our Privacy Policy

Reviewer #1: No

Reviewer #2: No

---

## [Author Response · Author response to Decision Letter 1]

27 May 2025

Please see the Response to Reviewers document.

---

## [Decision Letter · Decision Letter 1]

13 Jun 2025

Dear Dr. de Jong,

Thank you for submitting your manuscript to PLOS ONE. After careful consideration, we feel that it has merit but does not fully meet PLOS ONE’s publication criteria as it currently stands. Therefore, we invite you to submit a revised version of the manuscript that addresses the points raised during the review process.

We look forward to receiving your revised manuscript.

Kind regards,

Taher Babaee

Academic Editor

PLOS ONE

Journal Requirements:

Reviewers' comments:

Reviewer's Responses to Questions

**Comments to the Author**

Reviewer #1: (No Response)

Reviewer #2: All comments have been addressed

2. Is the manuscript technically sound, and do the data support the conclusions?

Reviewer #1: Yes

Reviewer #2: Yes

3. Has the statistical analysis been performed appropriately and rigorously?

Reviewer #1: Yes

Reviewer #2: Yes

4. Have the authors made all data underlying the findings in their manuscript fully available?

Reviewer #1: Yes

Reviewer #2: Yes

5. Is the manuscript presented in an intelligible fashion and written in standard English?

Reviewer #1: Yes

Reviewer #2: Yes

Reviewer #1: The authors are commended for substantially improving the methodological clarity and statistical reporting in this revised version. The multilevel modeling approach is appropriate, and diagnostic steps, robust estimation, and thoughtful interpretation are clearly described. The mixed-method design adds important contextual insight into patient experiences with the intervention. However, two key statistical issues remain insufficiently addressed in the current revision and should be resolved to ensure the manuscript meets the full transparency and rigor expected for publication.

Reporting and Handling of Missing Data

Although imputation methods for questionnaire subscales (e.g., WOMAC, SF-12) are described, the manuscript still does not present a clear account of missing data rates for the primary and secondary outcomes (e.g., VAS, 6-MWT) across all timepoints. Moreover, it is not explicitly stated whether all available outcome data were included in the multilevel models using full information maximum likelihood (FIML), which is standard under the missing at random (MAR) assumption. Please report the number of participants contributing data at each assessment timepoint for each outcome. Clearly state whether linear mixed models used all available repeated measures data under the MAR assumption, and confirm that no listwise deletion or case-wise exclusion was applied. If any dropout was non-random or outcome-related, briefly discuss the potential implications for bias.

Post Hoc Power or Precision Interpretation

The study acknowledges being underpowered relative to its initial target (n=80), yet the current version does not provide any post hoc calculation of detectable effect size or interpretive guidance based on the observed confidence intervals. This omission limits the ability to interpret the nonsignificant results for secondary outcomes and assess the likelihood of Type II error. Please include a brief post hoc calculation of the minimum detectable effect size (e.g., for VAS difference at 6 months) given the final sample size (n=23 per arm) and original assumptions. Alternatively, summarize the observed width of the 95% confidence intervals for key outcomes to contextualize the statistical precision and interpretability of the findings.

Reviewer #2: Reviewer Comments for Second Round of Review

The authors have adequately addressed most of the previous reviewers' comments with sound and scientific responses. However, the following minor revisions are recommended to further enhance the manuscript’s quality and transparency. Please address these points and submit the revised manuscript for final review.

1. Resolution of Figures (CONSORT Diagram)

Please confirm that the resolution of Figure 1 (CONSORT diagram) is at least 600 dpi and suitable for printing. If any issues with image clarity are identified, provide a higher-quality version.

2. Description of Biomechanical Mechanism of the Brace

In the Methods section (page 8, lines 157–159), add one or two sentences briefly describing the biomechanical mechanism of the SecuTec® OA brace (e.g., how it applies a valgus force to reduce load on the medial knee compartment). This will enhance replicability and reader understanding.

3. Reporting of Missing Data

Report the exact extent of missing data for each outcome (e.g., VAS, WOMAC, SF-12) in the main text or a supplementary table. This will increase transparency and allow readers to better assess the impact of missing data.

4. Integration of Quantitative and Qualitative Data

Add a brief paragraph in the Results or Discussion section to summarize the findings of S2 Table (triangulation of quantitative and qualitative data) and highlight how it informs result interpretation. Additionally, provide a simple comparative table to confirm that the characteristics of the interview subsample (e.g., age, gender, disease severity) do not significantly differ from the main sample.

5. Effect Size

Although your rationale for not reporting Cohen’s d in multilevel models is sound, please calculate and report an approximate effect size (e.g., Cohen’s d) for the primary outcome (VAS pain after 6-MWT) in the S1 File, accompanied by a brief explanation of its limitations in the context of multilevel models.

6. Units and Data Format Consistency

Review all tables and text to ensure that units (e.g., meters, degrees, centimeters) are clearly stated for all variables and that data formats (e.g., mean (SD) or n (%)) are consistent across all tables and text. Correct any inconsistencies identified.

7. Figure Legends

Expand the legend for Figure 2 (page 19, lines 325–339) to clearly explain the meaning of black and gray dots/lines, the range of scores (e.g., VAS, SF-12, WOMAC), and any other relevant details. This will ensure the figure is fully self-explanatory.

8. Explanation of Sample Size Reduction

Although reasons for participant exclusion post-randomization are provided in Figure 1 (page 7, lines 134–149), please add a brief sentence in the Discussion section (near lines 707–710, page 36) to clarify the practical reasons for the reduced sample size (e.g., logistical constraints or COVID-19 impact). This will provide additional context for readers.

9. Discussion of Disease Severity (Kellgren & Lawrence Grade)

The Results section (page 21, lines 376–379) notes that a higher percentage of participants with Kellgren & Lawrence Grade 3 in the intervention group underwent TKA or UKA post-study. Please add a brief sentence in the Discussion section to address this observation, noting its potential impact on brace effectiveness as a limitation or direction for future research.

10. Clarification of Qualitative Analysis Approach

In the Methods section (page 23, lines 271–290), add a brief sentence explaining why a deductive approach was chosen over an inductive approach for the qualitative analysis. This will help readers understand the methodological decision.

Overall Recommendation

The authors have addressed most of the previous comments effectively. Implementing the above minor revisions will further enhance the manuscript’s methodological rigor and clarity. The manuscript is expected to be suitable for acceptance following these changes. Please submit the revised version for final review.

**Do you want your identity to be public for this peer review?** For information about this choice, including consent withdrawal, please see our Privacy Policy

Reviewer #1: No

Reviewer #2: No

---

## [Author Response · Author response to Decision Letter 2]

24 Jun 2025

For a detailed response to all comments, please see Response to Reviewers_R2 document.

---

## [Decision Letter · Decision Letter 2]

29 Jul 2025

Effectiveness and user experiences of a valgus brace in patients with knee osteoarthritis: A mixed-method randomised controlled trial.

PONE-D-25-04724R2

Dear Dr. Jong,

We’re pleased to inform you that your manuscript has been judged scientifically suitable for publication and will be formally accepted for publication once it meets all outstanding technical requirements.

Kind regards,

Taher Babaee

Academic Editor

PLOS ONE

Additional Editor Comments (optional):

Reviewers' comments:

Reviewer's Responses to Questions

**Comments to the Author**

Reviewer #1: All comments have been addressed

2. Is the manuscript technically sound, and do the data support the conclusions?

Reviewer #1: Yes

3. Has the statistical analysis been performed appropriately and rigorously?

Reviewer #1: Yes

4. Have the authors made all data underlying the findings in their manuscript fully available?

Reviewer #1: Yes

5. Is the manuscript presented in an intelligible fashion and written in standard English?

Reviewer #1: Yes

Reviewer #1: All of my comments have been addressed very well. I have not further questions and I recommend that the journal accept it.

**Do you want your identity to be public for this peer review?** For information about this choice, including consent withdrawal, please see our Privacy Policy

Reviewer #1: No

---

## [Editor Report · Acceptance letter]

PONE-D-25-04724R2

PLOS ONE

Dear Dr. de Jong,

I'm pleased to inform you that your manuscript has been deemed suitable for publication in PLOS ONE. Congratulations! Your manuscript is now being handed over to our production team.

Kind regards,

on behalf of

Dr. Taher Babaee

Academic Editor

PLOS ONE